# Effects of Saffron Extract on Sleep Quality: A Randomized Double-Blind Controlled Clinical Trial

**DOI:** 10.3390/nu13051473

**Published:** 2021-04-27

**Authors:** Barbara D. Pachikian, Sylvie Copine, Marlène Suchareau, Louise Deldicque

**Affiliations:** 1Center of Investigation in Clinical Nutrition, Université catholique de Louvain, Place Pierre de Coubertin 1, Bte L8.10.02, 1348 Louvain-la-Neuve, Belgium; cicn@uclouvain.be (B.D.P.); sylvie.copine@uclouvain.be (S.C.); 2Comercial Quimica Masso, Viladomat, 321, 5º-08029 Barcelona, Spain; masso@cqmasso.com; 3Institute of Neuroscience, Université catholique de Louvain, Place Pierre de Coubertin 1, Bte L8.10.01, 1348 Louvain-la-Neuve, Belgium

**Keywords:** insomnia, actigraphy, LSEQ, PSQI, SF-36, quality of life, sleep disorder, crocins, safranal, crocetin

## Abstract

A saffron extract has been found to be effective in the context of depression and anxiety, but its effect on sleep quality has not been investigating yet using objective approaches. For this purpose, a randomized double-blind controlled study was conducted in subjects presenting mild to moderate sleep disorder associated with anxiety. Sixty-six subjects were randomized and supplemented with a placebo (maltodextrin) or a saffron extract (15.5 mg per day) for 6 weeks. Actigraphy was used to collect objective data related to sleep quality at baseline, at the middle and at the end of the intervention. Sleep quality was also assessed by completion of the LSEQ and PSQI questionnaires and quality of life by completion of the SF-36 questionnaire. Six weeks of saffron supplementation led to an increased time in bed assessed by actigraphy, to an improved ease of getting to sleep evaluated by the LSEQ questionnaire and to an improved sleep quality, sleep latency, sleep duration, and global scores evaluated by the PSQI questionnaire, whereas those parameters were not modified by the placebo. In conclusion, those results suggest that a saffron extract could be a natural and safe nutritional strategy to improve sleep duration and quality.

## 1. Introduction

Insomnia and sleep disorders are major concerns around the world, principally in western societies. Over a year, the prevalence of general sleep disturbance is estimated at approximately 85% and the diagnosis of primary insomnia estimated at around 10% [1]. In addition to generate problems during daily activities, these disorders can induce fatigue, attention deficits, mood instability, anxiety, or even depression [2,3]. Moreover, insomnia is associated to a reduction of the hippocampal volume, daytime cortical gamma-aminobutyric acid (GABA) levels, and recruitment of the caudate nucleus [4]. In several parts of the brain regions, like the hippocampus or prefrontal cortex, an inhibition of the usual activity decline occurring in the waking to sleep switch is observed. Thus, a general overactivity of the arousal, emotion regulating, and cognitive systems probably lead to the pathophysiology of insomnia [4]. To treat this disease, efficient drugs exist, like benzodiazepine or benzodiazepine-receptor agonists, but the use of those sleeping pills is restricted by their tolerance and an elevated risk of dependency, morbidity, and mortality with long-term use [4]. In this context, the search for safe and efficient compounds without adverse effects is essential.

Saffron, dried stigmas of the plant *Crocus sativus* L. used in traditional medicine, and its compounds, safranal, crocins, and crocetin have been largely studied for their effects on depression and anxiety in humans [5]. Additionally, saffron seems to induce beneficial effects on sleep duration and quality, as evidenced by previous reports using different approaches and protocols. Some studies used a whole saffron extract [6,7,8,9], others tested the effects of its active compound crocetin [10,11]. Different tools were used to assess sleep duration and quality. Objective measurements, such as electroencephalogram [11] or actigraphy [10] recordings, were used in only two studies, while all other studies opted for a subjective assessment of sleep related parameters using validated questionnaires. Finally, different doses were used, 7.5 mg per day for crocetin [10,11] and from 22 mg [6] to 300 mg [9] per day for the saffron extracts, as well as different durations of supplementation, from 1 week [9] to 8 weeks [8].

Except one study reporting no effects of 22 and 28 mg of a saffron extract for 4 weeks [6], all others found beneficial effects of saffron on sleep duration and/or quality. However, most of the data have been acquired based on questionnaires solely [6,7,8,9,12] and/or isolated purified molecules [10,11]. Isolated compounds rarely have the same degree of activity as the unrefined extract at comparable concentrations or dose of the active compound [13]. It is thus necessary to determine the effects of a saffron extract on sleep disorders with more objective tools, i.e., electronical systems. In consequence, the aim of the present study was to evaluate the effect of a standardized saffron extract on sleep quality using both questionnaires and actimeters allowing the measurement of the intensity, the amount, and the duration of physical movement in all directions.

## 2. Materials and Methods

### 2.1. Design

This interventional study is a double-blind, randomized, placebo-controlled parallel study. This study was approved by the local ethical committee (Comité d’Ethique Hospitalo-Facultaire UCLouvain/Cliniques Universitaires Saint-Luc). The trial was carried out in accordance with the Declaration of Helsinki and the Good Clinical Practice as required by the following regulations: the Belgian law of 7 May 2004 regarding experiments in human beings and the EU Directive 2001/20/EC on Clinical Trials (registration at clinicaltrials.gov as NCT04750681).

### 2.2. Study Population

Sixty-six volunteers were enrolled by the Center of Investigation in Clinical Nutrition (CICN, Belgium) between August 2019 and October 2020. They were recruited by posters, mails, social networks, and local newspapers. The screening visit, comprising a physical examination and blood sampling, was organized within 4 weeks before subject inclusion. Written informed consent was obtained from all subjects. To be included, the subjects had to meet the following criteria: woman or man aged between 25 and 70 years, presenting mild to moderate chronic primary sleep disorder (Insomnia Severity Index between 7 and 21), presenting mild to moderate anxiety (Perceived Stress Scale between 6 and 29), and for concerned woman use of effective contraception. The subjects were excluded if they presented one of the following exclusion criteria: a sleep disorder secondary to another health problem, a pharmacological resistance to common hypnotic drugs, a depressive disorder (Beck self-questionnaire score above 30), consumption of hypnotic drugs (<3 months before inclusion), gastro-intestinal, hepatic, respiratory, psychiatric, kidney, or cardiovascular disorder (<3 months before inclusion), abnormal blood sampling, recent (<3 months before inclusion) change in lifestyle (food, body weight, sport, drug, and/or dietary supplement), addiction or history of addiction, consumption of more than 3 glasses of alcohol per day, exaggerated consumption of tea (≥500 mL per day), coffee (≥400 mL per day), or energy drink (≥250 mL per day), pregnant or lactating woman, lifestyle habits which would modify the wake-sleep rhythm or which was expected to be modified during the study period (e.g., night work), and finally, a known allergy to saffron or olives.

### 2.3. Intervention

The subjects were randomly assigned to the placebo or the saffron group (stratified by gender). Both groups were instructed to ingest one capsule with a glass of water every day in the evening for 6 weeks. For the placebo group the capsule contained 275 mg of maltodextrin, whereas for the saffron group, the capsule contained 15.5 mg of a saffron extract (Saffr’activ^®^ SAF 3C PIM, Saffr’activ: Comercial Quimica Masso, Lyon, France; 1.6 mg of dry saffron extract, of which 0.9 mg of crocins and 0.7 mg of safranal) and 259.5 mg of maltodextrin. Chlorophyll capsules were used to mask the visual difference between placebo and saffron extract (capsule size T1 composed of hypromellose, water, and 0.7% chlorophyllin; K-caps^®^, CapsCanada^®^, Dania Beach, FL, USA). Sleep quality was assessed using objective (actigraphy) and subjective (questionnaires) approaches. One week before the intervention (baseline, between day -7 and day 0), at the middle of the intervention (week 3, between day 14 and day 21), and at the end of the intervention (week 6, between day 35 and day 42). A schematic of the study design is presented in Figure 1.

Protocol deviations and subjects’ withdrawals were monitored regularly during the study. The database was locked and the blinding was broken after completion of the whole quality control of the data.

### 2.4. Actigraphy

Actigraphy provides a non-invasive method to assess sleep-wake cycles over long periods, in the natural environment of the user [14]. One week before the intervention, the subjects wore an actimeter (MotionWatch 8, CamNtech Ltd., Cambridgeshire, UK). Participants were asked to activate the marker function on the watch when getting into bed and when rising the following morning. Moreover, they were asked to register these timings in a sleep diary. Two additional periods of actigraphy analysis were performed: one at the middle of the intervention (week 3, between day 14 and day 21); and the other at the end of the intervention (week 6, between day 35 and day 42). The following parameters were analyzed using the MotionWare 1.2.28 software (CamNtech Ltd., Cambridgeshire, UK): sleep efficiency (SE), sleep onset latency (SOL), time in bed (TIB), fragmentation index (FRAGI), total sleep time (TST), and wake after sleep onset (WASO). Data were averaged for each one-week period.

### 2.5. Questionnaires

The Leeds Sleep Evaluation Questionnaire (LSEQ) [15] was completed at day −7 and day 0 (at baseline, before the intervention), at the middle of the intervention (day 21), and at the end of the intervention (day 42). The questionnaire is a standardized self-reporting instrument comprising ten visual analogue scales (10 cm) to assess sleep quality. Four items were calculated: “the ease to getting to sleep”, “the quality of sleep”, “the ease of awakening from sleep”, and finally “the alertness and behavior following wakefulness”. Each item ranges from 0 to 100, the higher the score, the better the sleep quality. In addition, the Pittsburgh Sleep Quality Index (PSQI) [16] and short-form 36 items (SF-36, developed at RAND as part of the Medical Outcomes Study) questionnaires were completed at day −7 and day 0 (at baseline, before the intervention), and at the end of the intervention (day 42). The PSQI is a self-reporting questionnaire with 19 items that assesses sleep quality over a 1-month period. The 19 items are divided into 7 components that measure subjective sleep quality, sleep latency, sleep duration, sleep efficiency, sleep disturbances, use of sleeping medications, and daytime dysfunction. Each seven components are scored between 0 and 3 and the global PSQI score is calculated as the sum of the 7 components’ scores. A score of 0 would mean an absence of sleep difficulties and a score of 21 would mean major sleep difficulties. RAND SF-36 is a self-reporting questionnaire comprising 36 items to assess quality of life. Each item is scored on a scale from 0 to 100. Eight components were calculated to measure physical functioning, physical limitations, bodily pain, general health, vitality, social functioning, emotional limitations, and mental health. A higher score would mean a better quality of life.

### 2.6. Follow-Up

All un-used capsules were brought back to the investigation center at the end of the intervention. The compliance was calculated as the number of capsules consumed divided by the number of capsules that was indicated based on the duration of the supplementation and expressed as a percentage.

Any adverse event and/or concomitant medication reported by a participant was registered. Food diaries were completed three days (two weekdays and one weekend day) at baseline and at the end of the intervention to ensure the absence of important dietary changes.

### 2.7. Statistical Analysis

Sample size was calculated based on the primary endpoint, i.e., changes in the perceived quality of sleep assessed by the LSEQ questionnaire. Using the software PASS 14.0.7 and the one-sided Two-Sample T-test (assuming equal variance), 64 subjects were needed (with estimated 10% drop out subjects) to observe a difference of 10 points between the LSEQ means scores of the test and control groups. The standard deviation was set at 15, power at 80%, and alpha at 0.05.

Statistical analyses were performed using the software systems SAS 9.4 (SAS Institute Inc., Cary, NC, USA) on the per-protocol population. The evolution of an endpoint between the baseline and week 3 and/or week 6 within a group was evaluated using the non-parametric Wilcoxon signed-rank Test. This test allows to determine the intervention effect inside a group. The endpoint changes from baseline at each intervention period (week 3-baseline or week 6-baseline) were compared between the placebo and test groups using the non-parametric Mann–Whitney two-sample test. This test allows to determine the intervention effect between groups, considering the baseline. Data are shown as mean with SD. All analyses were conducted at an alpha level of 0.05.

## 3. Results

### 3.1. Baseline Characteristics

Sixty-six subjects were randomized into the placebo (*n* = 32) or the saffron (*n* = 34) group (Figure 2). In the placebo group, one subject did not receive the treatment because of the use of a sleeping drug before the beginning of the intervention and one subject was excluded from the analyses because of the lack of actimeter data for the last week of recording (week 6). In the saffron group, one subject discontinued the intervention because of an adverse event maybe linked to the product and two others because of stressful situations (at work and in family), which led to the use of sleeping drugs. Two subjects were excluded from the analyses; one because of the use of sleeping medication and the other because the last week of actimeter data recording was done with a delay of 10 days.

Table 1 presents the baseline characteristics of the enrolled participants. A high compliance was evaluated with an intake of 97.4 ± 7.2% (*n* = 28) in the placebo group and 96.3 ± 8.2% (*n* = 31) in the saffron group. Using food diaries, we observed no changes in total energy, protein, alcohol, tea, or soda intake during the intervention in both groups (data not shown). During the 6-week intervention, carbohydrate intake increased in both the placebo (% of total energy intake, 43.5 ± 5.7 versus 41.5 ± 7.2, *p* = 0.030) and the saffron group (% of total energy intake, 42.8 ± 7.8 versus 38.9 ± 10.1, *p* = 0.010), and coffee intake increased only in the saffron group (mL/day, 172.4 ± 200.1 versus 149.9 ± 167.5, *p* = 0.030).

### 3.2. Sleep Quality

#### 3.2.1. Actigraphy

No difference in the actimeter data was measured after 6 weeks of intervention compared to the baseline in the placebo group (Table 2). In the saffron group, only TIB tended to be higher after 6 weeks of intervention compared to the baseline (*p* = 0.051). The changes in TIB between the baseline and 6 weeks was different between the saffron and the placebo group, with the saffron group increasing TIB but not the placebo group (*p* = 0.023).

#### 3.2.2. Questionnaires

Sleep quality was evaluated thanks to two different questionnaires. Regarding the LSEQ scores, an increase in the “quality of sleep”, “ease of awakening from sleep”, and “alertness and behavior following wakefulness” scores were observed in both the placebo and saffron groups after 3 and 6 weeks compared to baseline (*p* < 0.05), with no difference between placebo and saffron (Table 3). For the “ease of getting to sleep”, only the saffron group exhibited a higher score after 6 weeks of intervention compared to the baseline (*p* = 0.041).

For the PSQI scores, compared to baseline, 6 weeks of saffron supplementation, but not placebo, led to a lower score in sleep quality (*p* = 0.014), sleep latency (*p* = 0.032), and sleep duration (*p* = 0.013) (Table 4). For the latter, the change from baseline was significantly different from the placebo group (*p* = 0.021). The daytime dysfunction score was decreased after 6 weeks of intervention in both the placebo (*p* < 0.001) and saffron (*p* < 0.001) groups, whereas all other scores remained unchanged. The global PSQI score decreased after 6 weeks of saffron supplementation, but not placebo, compared to baseline (*p* = 0.001).

### 3.3. Quality of Life

Based on the SF-36 scores, six weeks of saffron supplementation led to an improvement of the global physical score (*p* = 0.041), mainly due to an improvement of the bodily pain score (*p* = 0.035) and a trend to an increase in the physical functioning (*p* = 0.055) and general health score (*p* = 0.053) (Table 5). The placebo had no effect on those scores related to the physical well-being. Both placebo and saffron led to an improvement of the vitality score, the mental health score, and the global mentality score compared to baseline (*p* < 0.05). The social functioning score was improved after 6 weeks of intervention in the placebo group only (*p* = 0.002). The changes in the social functioning score between the baseline and 6 weeks was different between the saffron and the placebo group, with the placebo group decreasing this score but not the saffron group (*p* = 0.038). Finally, only saffron supplementation led to an improvement of the emotional limitation score compared to the baseline (*p* = 0.005).

### 3.4. Safety

The saffron supplementation was well tolerated, with no serious adverse event reported and no difference in the mild and moderate adverse events reported by the placebo (*n* = 18) and the saffron (*n* = 21) groups. One mild adverse event was reported in the saffron group, i.e., palpitations in the evening after the consumption of the product. The subject was excluded from the study and the palpitations stopped immediately after the interruption of the intervention.

## 4. Discussion

This study aimed to evaluate the effect of a standardized saffron extract (15.5 mg per day for 6 weeks) on sleep quality of subjects presenting mild to moderate sleep troubles associated to anxiety. We observed that 6 weeks of saffron extract supplementation led to an improvement of several parameters related to sleep quality: TIB evaluated by actigraphy, the ease of getting to sleep evaluated by the LSEQ questionnaire, the sleep quality, sleep latency, sleep duration, and global scores evaluated by the PSQI questionnaire. In addition, saffron supplementation improved the bodily pain score, the physical global score, and the emotional limitation score assessed by the SF-36 questionnaire.

To the best of our knowledge, only one study looked at the effects of a saffron extract on sleep related parameters in participants suffering from mild to moderate chronic primary sleep disorder based on the insomnia severity index [7]. The other studies investigated the effects of crocetin [10,11] or type 2 diabetic patients [8,9] or depressed adults [6]. Our results confirm and complete the study of Lopresti et al., which evaluated the effects of a 28-day saffron extract supplementation (2 × 14 mg per day) in subjects with mild to moderate sleep disorders [7]. Compared to placebo, saffron led to a greater reduction of the insomnia severity score (without, however, eliminating the problem of insomnia) and the non-restorative sleep score measured by the Relationship Scales Questionnaire (RSQ). Based on a sleep diary, saffron increased the rating of sleep quality. In the latter study, the decrease in the ISI severity score was observed rapidly, i.e., after 7 days of supplementation [7]. Here, the effects of the saffron extract on sleep related parameters were detectable only after 6 weeks, with a tendency for some parameters to be improved after 3 weeks of intervention. In both studies, the saffron extract contained crocins and safranal, however in different quantities. In Lopresti et al., the extract contained at least 3.5% of active compounds, which represents 0.980 mg per day. In the present study, the participants received 0.9 mg crocins and 0.7 mg safranal per day, totalizing 1.6 mg of active compounds per day. It is therefore difficult to explain the earlier effect on the ISI score in Lopresti et al. by a higher active compound content in their extract, on the contrary. Anyway, despite this difference in the timing, both studies observed a positive effect of a saffron extract on the ISI score, which is eventually an important effect of saffron related to sleep.

Compared to the study of Lopresti et al. [7], based only on questionnaires, the added value of the present study is the use of actimetry, an objective approach to study sleep quality. Using actigraphy, we found that our saffron extract increased TIB. Though, the accepted gold standard for sleep assessment is polysomnography. This technique requires that participants either come to a sleep laboratory or be connected to portable polysomnography equipment at home, creating a considerable burden to participants. Moreover, because of the high variability of insomniac sleep and bias related to the first night effect, polysomnography is not indicated for routine evaluation of chronic sleep trouble. For all those reasons, we chose to use actigraphy, which is considered to be a reliable alternative to polysomnography for monitoring treatment response among patients with insomnia and to compare pre- and post-treatment sleep quality [17]. In addition, actigraphy is less intrusive, subjects can be studied in their own home environment for multiple nights and is relatively easy to use in ambulatory settings [18]. It should be pointed out that our saffron extract supplementation increased TIB by 16 min compared to baseline and by 25 min compared to placebo. Compared to chemical sleeping drugs, those improvements should be considered as modest but still interesting when compared to other dietary interventions and/or herbal products, based on actigraphy data as well [19,20].

Though not directly comparable as different populations were tested than in the present study, other studies found a positive effect of a saffron extract or an isolated saffron active compound on sleep parameters. In 21 healthy adults, a 4-week saffron extract supplementation led to an improvement of the PSQI global score compared to placebo [12]. In 50 diabetic subjects, a 1-week saffron supplementation (300 mg per day) improved ease of getting to sleep, sleep duration, sleep efficiency, sleep disturbances, daytime function, and total PSQI scores compared to placebo [9]. In 54 diabetic subjects suffering from anxiety and depression, an 8-week saffron extract supplementation (30 mg per day) led to improved PSQI global score compared to placebo [8]. The supplementation with 7.5 mg per day of crocetin for 2 weeks, in a randomized crossover study with 24 subjects presenting mild sleep complaints, improved the delta power measured by electroencephalography and two subjective parameters: the sleepiness on rising and the feeling refreshed, assessed by the OSA-MA questionnaire, compared to placebo [11]. In a similar design in 21 subjects with mild sleep disorder (PSQI > 5), a supplementation with 7.5 mg per day of crocetin during 2 weeks improved the maintaining of sleep assessed by actigraphy, whereas no effect on subjective parameters was reported (SMHSQ questionnaire) [10]. Though, it should be reported as well that one study in 128 healthy adults with self-reported low mood found no effect of 22 or 28 mg per day of a saffron extract for 4 weeks on the PSQI score compared to placebo [6]. Altogether, with the exception of the latter study, those results indicate that several saffron extracts as well as crocetin have a measurable positive effect on sleep duration and sleep quality.

The mechanisms of action of saffron extract on sleep quality and sleep duration is not totally known. Looking at the effects of the active compounds of saffron, safranal was found to activate the sleep-promoting neurons from the ventrolateral preoptic nucleus (VLPO) and inhibit the wakefulness-promoting neurons from the tuberomammillary nuclei (TMN) in vitro [21]. In mice, crocins were suggested to modulate the histaminergic or cholinergic arousal system to induce non-REM sleep [22]. Moreover, saffron extract and its active compounds are well known to modulate the levels of serotonin, dopamine, norepinephrine, glutamate, and GABA-A neurotransmitters [9,23,24]. Based on their role in anxiety, depression, and insomnia [2,25,26,27,28], the regulation of the brain levels of those neurotransmitters by saffron probably largely contribute to the positive effects of saffron on sleep quality.

In addition to an improvement in sleep duration and ease of getting to sleep, we found beneficial effects of saffron on quality of life parameters. Our 6-week saffron extract supplementation reduced pain feeling, improved physical global score, and reduced emotional limitation, assessed by the SF-36 questionnaire. Of note, the saffron baseline values of physical functioning, bodily pain, and general health were slightly lower than the placebo baseline values, resulting in a lower, but non-significantly different, saffron baseline physical global score compared to placebo. Similarly, the social functioning score improved only in the placebo group. However, the placebo baseline score was lower, but once again non-significantly different, than the saffron baseline score. The improvements observed in those scores should thus be interpreted with caution, due to slight, but non-significant, differences at baseline. A few studies previously looked at the effect of saffron on quality of life parameters, with divergent results. In depressive patients supplemented with both an antidepressant and 28 mg saffron extract for 8 weeks, no effect could be observed on the different SF-36 scores compared to the placebo [29]. In other contexts, a saffron extract supplementation was found to reduce pain in patients suffering from rheumatoid arthritis [30] or to reduce the delayed-onset muscle soreness after eccentric exercise in university students [31]. Altogether those results suggest that saffron could be interesting in other contexts than anxiety and depression [32,33] but further studies are needed to confirm those effects.

Our saffron supplementation was safe, as no differences in mild and moderate adverse events were reported between both groups. One subject in the saffron group experienced palpitations after product ingestion in the evening. Palpitations stopped directly after intervention.

Several limitations of this study need to be considered. The dosage of saffron extract (15.5 mg per day) in this study was based on the standardization of crocins (0.9 mg per day) and safranal (0.7 mg per day). The efficacity and safety of 30 mg of saffron extract commonly used to improve depression and anxiety [8,34,35], with standardization of crocins ≥ 0.9 mg and standardization of safranal ≥ 0.6 mg require further investigation. Moreover, the population studied presented mild to moderate chronic primary sleep disorder, the effects of saffron extract on populations with severe sleep disorder is unknown.

In conclusion, the present study provides additional evidence that a saffron extract could be an interesting natural and safe nutritional strategy to improve sleep duration and quality of sleep on population presented mild to moderate chronic primary sleep disorder.

## Figures and Tables

**Figure 1 nutrients-13-01473-f001:**
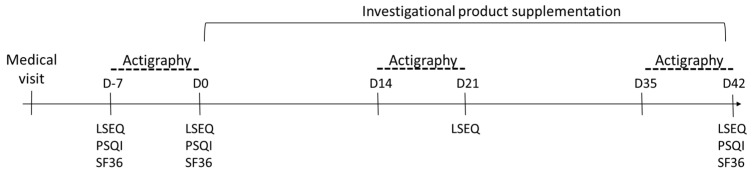
Schematic representation of the study design.

**Figure 2 nutrients-13-01473-f002:**
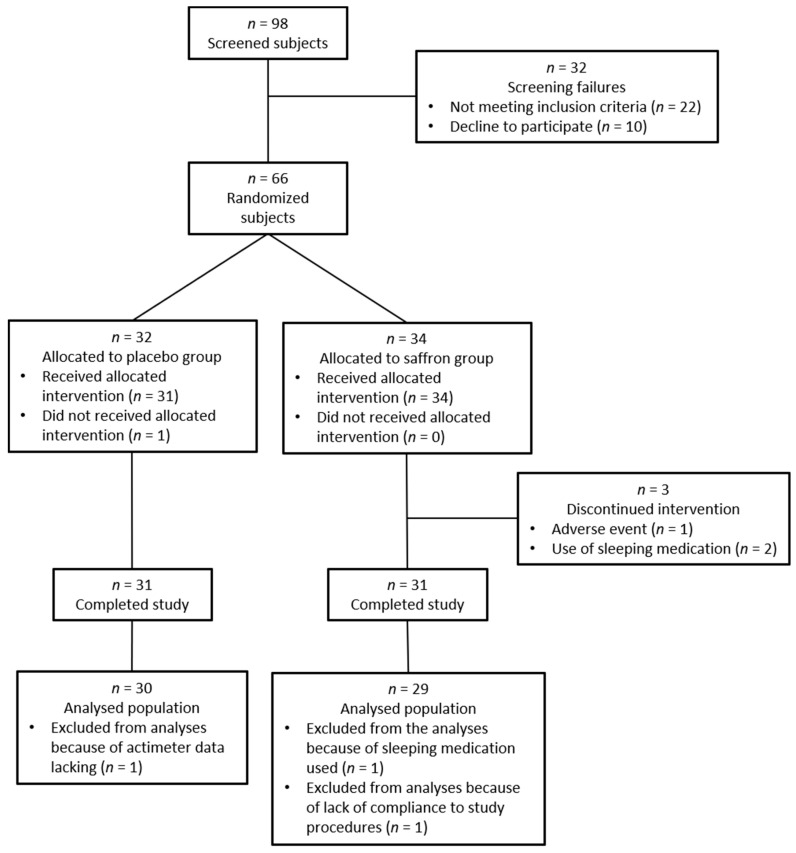
CONSORT flow diagram.

**Table 1 nutrients-13-01473-t001:** Characteristics of the study per-protocol participants at baseline (*n* = 59).

		Placebo (*n* = 30)	Saffron (*n* = 29)
		Baseline	
Age (years)	Mean (SD)	44 (15)	46(13)
Men	n (%)	10 (17%)	11 (19%)
Women	n (%)	20 (34%)	18 (30%)
Independent	n (%)	2 (3%)	1 (2%)
Employee	n (%)	23 (39%)	24 (41%)
Inactive	n (%)	3 (5%)	3 (5%)
Student	n (%)	2 (3%)	1 (2%)
BMI (kg/m^2^)	Mean (SD)	24.3 (4.0)	26.2 (5.1)
Systolic blood pressure (mmHg)	Mean (SD)	126 (15)	131 (15)
Diastolic blood pressure (mmHg)	Mean (SD)	80 (11)	82 (10)
PSS score at V0	Mean (SD)	19.6 (4.8)	18.9 (4.8)
ISI score at V0	Mean (SD)	14.0 (3.3)	13.1 (3.3)
BDI score at V0	Mean (SD)	12.9 (5.8)	12.2 (7.0)
Not a sleeping medication user	n (%)	25 (42%)	27 (46%)
Sleeping medication user	n (%)	5 (9%)	2 (3%)

PSS perceived stress scale, ISI insomnia severity index, BDI Beck self-questionnaire score.

**Table 2 nutrients-13-01473-t002:** Actigraphy variables.

		Placebo			Saffron		
		Baseline (*n* = 30)	Week 3 (*n* = 30)	Week 6 (*n* = 30)	Baseline (*n* = 29)	Week 3 (*n* = 29)	Week 6 (*n* = 29)
TIB (min)	Mean (SD)	484.5 (41.9)	477.9 (50.2)	475.8 (44.9)	486.6 (44.9)	491.7 (42.0)	500.7 (41.4) ^$^
TST (min)	Mean (SD)	387.8 (27.3)	381.6 (34.6)	379.7 (34.9)	368.8 (48.3)	370.4 (41.3)	376.7 (42.5)
SOL (min)	Mean (SD)	9.6 (11.4)	8.2 (7.9)	7.5 (6.3)	16.4 (18.9)	18.2 (19.3)	14.7 (16.1)
WASO (min)	Mean (SD)	83.4 (25.8)	82.6 (28.4)	81.8 (26.6)	97.7 (30.7)	97.3 (33.3)	103.3 (30.5)
FRAGI	Mean (SD)	27.3 (9.5)	26.9 (10.7)	26.7 (9.0)	32.6 (12.5)	32.0 (13.7)	33.9 (13.4)
SE (%)	Mean (SD)	80.4 (5.4)	80.3 (5.6)	80.2 (5.7)	76.0 (8.2)	75.6 (8.0)	75.4 (7.8)

TIB time in bed, TST total sleep time, SOL sleep onset latency, WASO wake after sleep onset, FRAGI fragmentation index, SE sleep efficiency. ^$^ endpoint changes from the baseline significantly different between group, *p* < 0.05 (Mann-Whitney U test).

**Table 3 nutrients-13-01473-t003:** LSEQ scores.

		Placebo			Saffron		
		Baseline (*n* = 30)	Week 3 (*n* = 30)	Week 6 (*n* = 30)	Baseline (*n* = 29)	Week 3 (*n* = 29)	Week 6 (*n* = 29)
Ease of getting to sleep	Mean (SD)	51.4 (8.5)	55.0 (11.2)	53.4 (10.8)	47.1 (7.3)	52.1 (8.5)	52.4 (7.2) *
Quality of sleep	Mean (SD)	40.3 (11.2)	52.1 (14.1) *	49.1 (16.9) *	42.3 (14.2)	53.9 (10.5) *	51.5 (12.6) *
Ease of awakening from sleep	Mean (SD)	44.0 (13.0)	49.5 (10.6) *	51.2 (15.2) *	44.7 (10.6)	49.2 (11.7) *	51.9 (12.7) *
Alertness and behavior following wakefulness	Mean (SD)	39.6 (14.0)	49.4 (13.5) *	47.7 (15.9) *	40.0 (11.2) *	51.7 (15.5) *	51.8 (17.2) *

* significantly different from baseline within a group, *p* < 0.05 (Wilcoxon signed-rank test).

**Table 4 nutrients-13-01473-t004:** PSQI scores.

		Placebo		Saffron	
		Baseline (*n* = 30)	Week 6 (*n* = 30)	Baseline (*n* = 29)	Week 6 (*n* = 29)
Sleep quality	Mean (SD)	1.65 (0.51)	1.40 (0.77)	1.83 (0.47)	1.55 (0.63) *
Sleep latency	Mean (SD)	1.50 (0.86)	1.23 (1.35)	1.45 (0.85)	1.18 (0.82) *
Sleep duration	Mean (SD)	0.80 (0.66)	0.87 (0.68)	1.12 (0.90)	0.71 (0.76) *^,$^
Sleep efficiency	Mean (SD)	0.63 (0.61)	0.83 (0.87)	0.84 (1.02)	0.64 (0.91)
Sleep disturbances	Mean (SD)	1.43 (0.45)	1.33 (0.48)	1.48 (0.47)	1.38 (1.00)
Use of sleeping medication	Mean (SD)	0.27 (0.65)	0.20 (0.76)	0.24 (0.79)	0.27 (0.80)
Daytime dysfunction	Mean (SD)	1.47 (0.67)	0.97 (0.56) *	1.34 (0.52)	0.76 (0.63) *
Global score	Mean (SD)	7.75 (1.84)	6.83 (2.64)	8.18 (2.36)	6.46 (2.66) *

* significantly different from the baseline within a group, *p* < 0.05 (Wilcoxon signed-rank test) and ^$^ endpoint changes from the baseline significantly different between group, *p* < 0.05 (Mann–Whitney U test).

**Table 5 nutrients-13-01473-t005:** SF-36 scores.

		Placebo		Saffron	
		Baseline (*n* = 30)	Week 6 (*n* = 30)	Baseline (*n* = 29)	Week 6 (*n* = 29)
Physical functioning	Mean (SD)	86.58 (14.70)	86.50 (17.33)	83.19 (16.46)	87.07 (15.56)
Physical limitation	Mean (SD)	62.09 (32.58)	75.83 (34.42)	68.96 (29.43)	75.00 (34.72)
Bodily pain	Mean (SD)	70.62 (19.64)	70.75 (25.71)	69.40 (20.29)	74.74 (21.97) *
General health	Mean (SD)	69.03 (13.89)	68.33 (19.28)	64.22 (13.95)	67.82 (15.59)
Physical global score	Mean (SD)	72.08 (15.98)	75.35 (20.22)	71.44 (16.24)	76.16 (18.02) *
Vitality	Mean (SD)	45.42 (16.51)	51.17 (17.75) *	46.72 (14.79)	53.27 (17.07) *
Social functioning	Mean (SD)	60.85 (17.05)	71.32 (22.30) *^,$^	68.85 (15.59)	71.90 (17.21)
Emotional limitation	Mean (SD)	56.67 (33.79)	67.78 (34.44)	60.92 (34.00)	78.16 (33.66) *
Mental health	Mean (SD)	56.80 (16.97)	63.60 (20.05) *	56.62 (17.25)	62.90 (17.27) *
Mental global score	Mean (SD)	54.93 (17.68)	63.46 (59.06) *	58.28 (15.77)	66.56 (18.44) *

* significantly different from the baseline within a group, *p* < 0.05 (Wilcoxon signed-rank test) and ^$^ endpoint changes from the baseline significantly different between group, *p* < 0.05 (Mann–Whitney U test).

## Data Availability

Data are available upon request by sending an e-mail to masso@cqmasso.com.

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
