# Peer review of "Effects of Saffron Extract on Sleep Quality: A Randomized Double-Blind Controlled Clinical Trial"

_nutrients, 2021, doi:10.3390/nu13051473_

Round 1

Reviewer 1 Report

The prospective clinical study described in the manuscript nutrients-1170015 aims at characterising efficacy and safety of a saffron extract. The authors performed a randomised, placebo-controlled, double-blind clinical trial, which is the adequate design to address such goals. The used questionnaires are standard in the assessment of sleep quality. Moreover, the authors introduced actigraphy, to complement the questionnaire results, which is highly recommended. However there are several points that must be clarified before I can recommend publication of this manuscript.

MAJOR POINTS:

  • In the methods’ subsection “Intervention” more information on how the capsules were ingested should be added. Were they shewed, taken with some water? This information is highly important because of the next point.
  • The authors describe the study as double-blind, but the placebo control – maltodextrin – can in principle be easily distinguished from a mixture of saffron extract in maltodextrin in terms of colour and taste. If the authors took measures to make the placebo and verum indistinguishable, these should be explained. If it was somehow possible to distinguish between both, this is a strong study limitation and should be discussed as such (in the discussion, in the paragraph about study limitations).
  • The authors mention that the study was performed according to GCP, however there is no information on the independent monitoring that is inherent to GCP. Information on the monitoring should be added.
  • The authors write that results from an ITT-analysis are shown. However, in Fig. 2 it is apparent that 1 patient in the placebo group and 2 patients in the saffron group were not considered in the analysis. Why were patients excluded from the ITT analysis? Or are the results shown from a PP-analysis?
  • There are conflicting data on the number of patients: 66 were randomised (Fig. 2), demographic data are shown for 65 enrolled patients (Table 1), analysed were 59 (Fig. 2 and results’ tables). Please explain or correct.
  • It is unclear what was the primary endpoint of the study (of the numerous endpoints described). Please clarify.
  • In a placebo-controlled study, the emphasis lays on the comparison between placebo and verum and in most cases no difference was found. For instance, the SF-36 data on mental scales reveal that no difference between placebo and verum was attained. In fact, the only significant difference between placebo and verum concerns the social functioning. These aspects should be discussed.
  • The SF-36 physical functioning values at baseline are rather high, as to be expected from the inclusion criteria. The only value that is clearly different from the other 3 values (considering 2 groups, before and after) is the value of the saffron group at baseline (Table 5). The discussion on data on physical improvements should therefore be interpreted with caution and rephrased (this different value at baseline should be mentioned and no strong conclusions can be taken).
  • According to the authors (line 278) the added value of the present study is the actigraphy. However, the results show an improvement of only 14 min after 6 weeks treatments. This improvement has to be seen as very modest and discussed as such.

Author Response

We thank the reviewer for his/her useful comments regarding the manuscript evaluating the effects of saffron extract on sleep quality in the context of mild to moderate insomnia. Please find below our answers.

The prospective clinical study described in the manuscript nutrients-1170015 aims at characterising efficacy and safety of a saffron extract. The authors performed a randomised, placebo-controlled, double-blind clinical trial, which is the adequate design to address such goals. The used questionnaires are standard in the assessment of sleep quality. Moreover, the authors introduced actigraphy, to complement the questionnaire results, which is highly recommended. However there are several points that must be clarified before I can recommend publication of this manuscript.

MAJOR POINTS:

  • In the methods’ subsection “Intervention” more information on how the capsules were ingested should be added. Were they shewed, taken with some water? This information is highly important because of the next point.

Volunteers were instructed to swallow one capsule with a glass of water every evening. The capsules were not to be chewed. Precision were made in the manuscript line 98.

  • The authors describe the study as double-blind, but the placebo control – maltodextrin – can in principle be easily distinguished from a mixture of saffron extract in maltodextrin in terms of colour and taste. If the authors took measures to make the placebo and verum indistinguishable, these should be explained. If it was somehow possible to distinguish between both, this is a strong study limitation and should be discussed as such (in the discussion, in the paragraph about study limitations).

We used chlorophyll-based capsules (K-caps®) to mask the visual difference between placebo and saffron. Precision were made in the manuscript lines 102 to 104.

  • The authors mention that the study was performed according to GCP, however there is no information on the independent monitoring that is inherent to GCP. Information on the monitoring should be added.

As proposed in the GCP, the Sponsor should determine the appropriate extent of the monitoring. The initiation visit was realized at the beginning of the trial. Then, meetings were scheduled 4 times during the study between the Sponsor and the Investigator staff (about every 20 subjects). A final meeting was organized to close the trial. The following points were followed during each meeting: subjects’ withdrawals and protocol deviations. The database was locked and the blinding was broken after completion of the whole quality control of the data.  Precision were made in the manuscript lines 113 to 115.

  • The authors write that results from an ITT-analysis are shown. However, in Fig. 2 it is apparent that 1 patient in the placebo group and 2 patients in the saffron group were not considered in the analysis. Why were patients excluded from the ITT analysis? Or are the results shown from a PP-analysis?

Indeed, it was a mistake. Results were shown for the per protocol population. Correction was made line 168. The results were also analyzed on the Intention-to-treat population and were similar.

  • There are conflicting data on the number of patients: 66 were randomised (Fig. 2), demographic data are shown for 65 enrolled patients (Table 1), analysed were 59 (Fig. 2 and results’ tables). Please explain or correct.

We have now corrected Table 1 to include the per-protocol subjects (line 197 to 199). 66 subjects were randomized but one did not receive the allocated treatment. In fact, he passed the screening visit but just before the beginning of the study he started a treatment with trazodone for a cervical hernia and was therefore excluded. 59 is the number of subjects in the per protocol population. For information similar results were obtained for the ITT population analyses.

  • It is unclear what was the primary endpoint of the study (of the numerous endpoints described). Please clarify.

We added a sentence line 161 to 162 to clarify that the primary endpoint (which was used for the calculation of the sample size) is the change in LSEQ perceived quality of sleep score.

  • In a placebo-controlled study, the emphasis lays on the comparison between placebo and verum and in most cases no difference was found. For instance, the SF-36 data on mental scales reveal that no difference between placebo and verum was attained. In fact, the only significant difference between placebo and verum concerns the social functioning. These aspects should be discussed.

Social functioning was significantly improved compared to baseline in the placebo but not in the saffron group. However, as for the global physical score, the baseline value in the placebo was different between the placebo and saffron groups. At the end of the treatment saffron and placebo social functioning scores were similar. We modified the discussion related to that point (lines 338-345) to be more careful in the interpretation.

  • The SF-36 physical functioning values at baseline are rather high, as to be expected from the inclusion criteria. The only value that is clearly different from the other 3 values (considering 2 groups, before and after) is the value of the saffron group at baseline (Table 5). The discussion on data on physical improvements should therefore be interpreted with caution and rephrased (this different value at baseline should be mentioned and no strong conclusions can be taken).

Using Mann-Withney test, we confirmed that the baseline values for all SF36 subscores were not different between placebo and saffron. However, we changed the sentences in the discussion (lines 338-345 and 351-353) to be more careful in the interpretation.

  • According to the authors (line 278) the added value of the present study is the actigraphy. However, the results show an improvement of only 14 min after 6 weeks treatments. This improvement has to be seen as very modest and discussed as such.

We agree with the reviewer that dietary supplements may not be compared to sleeping drugs, which are more efficient but as mentioned in the introduction, also induce many side effects. The modest improvements in sleep duration compared to drugs is now discussed lines 297 to 302.

Reviewer 2 Report

The manuscript is well written and the experimental design consistent.

I propose minor revisions.

1- The authors must insert in the materials and methods paragraph a sub-paragraph with detailed chemical composition of the saffron extract and all the excipients present in the commercial product used, possibly adding a table and the chemical structures of the main secondary metabolites.

2- In addition, it is appropriate to indicate whether any cardiovascular problems of the patients were an element of exclusion from the study.

3- It is appropriate to detail in a sub-paragraph any cardiovascular effects manifested during the study in the light of the scientific literature that describes cardiovascular effects for saffron extract and the relationship that exists between these effects and the concentration of the secondary metabolites such as crocin and kaempferol present in the commercial product used.

Author Response

The manuscript is well written and the experimental design consistent.

I propose minor revisions.

1- The authors must insert in the materials and methods paragraph a sub-paragraph with detailed chemical composition of the saffron extract and all the excipients present in the commercial product used, possibly adding a table and the chemical structures of the main secondary metabolites.

The composition of the excipients was added in the material and method section (lines 101-104).

2- In addition, it is appropriate to indicate whether any cardiovascular problems of the patients were an element of exclusion from the study.

As described in the material and method section line 88, subjects with cardiovascular disorder were excluded from this study.

3- It is appropriate to detail in a sub-paragraph any cardiovascular effects manifested during the study in the light of the scientific literature that describes cardiovascular effects for saffron extract and the relationship that exists between these effects and the concentration of the secondary metabolites such as crocin and kaempferol present in the commercial product used.

Only one cardiovascular effect was reported during the study: palpitations in one subject from the saffron group, which stopped directly after the intervention. Still, saffron is generally recognized as beneficial in the context of cardiovascular diseases. In fact, saffron stigmas extract present anti-oxydant and anti-inflammatory activities and one study in rats described beneficial effects of a saffron extract in case of arrythmia (PMID: 23627471). We have now added a few sentences concerning the cardiovascular effect reported here in the discussion lines 354-357.

Kaempferol is an active compound mainly present in saffron petal. However, we worked exclusively with a stigmas saffron extract.

Round 2

Reviewer 1 Report

The revised version is superior to the first one.